# Isolated zero field sub-10 nm skyrmions in ultrathin Co films

Sebastian Meyer[1,3], Marco Perini[2,3], Stephan von Malottki[1], André Kubetzka[2], Roland Wiesendanger[2], Kirsten von Bergmann [2] & Stefan Heinze[1]

Due to their exceptional topological and dynamical properties magnetic skyrmions—localized stable spin structures—show great promise for spintronic applications. To become technologically competitive, isolated skyrmions with diameters below 10 nm stable at zero magnetic field and at room temperature are desired. Despite finding skyrmions in a wide spectrum of materials, the quest for a material with these envisioned properties is ongoing. Here we report zero field isolated skyrmions at $T = 4\,K$ with diameters below 5 nm observed in the virgin ferromagnetic state coexisting with 1 nm thin domain walls in Rh/Co atomic bilayers on Ir(111). These spin structures are investigated by spin-polarized scanning tunneling microscopy and can also be detected using non-spin-polarized tips via the noncollinear magnetoresistance. We demonstrate that sub-10 nm skyrmions are stabilized in these ferromagnetic Co films at zero field due to strong frustration of exchange interaction, together with Dzyaloshinskii–Moriya interaction and large magnetocrystalline anisotropy.

[1] Institute of Theoretical Physics and Astrophysics, Christian-Albrechts-Universität zu Kiel, Leibnizstrasse 15, 24098 Kiel, Germany. [2] Department of Physics, University of Hamburg, Jungiusstrasse 11, 20355 Hamburg, Germany. [3] These authors contributed equally: Sebastian Meyer, Marco Perini. Correspondence and requests for materials should be addressed to K.v.B. (email: kbergman@physnet.uni-hamburg.de)

The stabilization of isolated magnetic skyrmions[1] at zero magnetic field in a ferromagnetic (FM) material due to the Dzyaloshinskii–Moriya interaction (DMI)[2,3] has been predicted already more than 20 years ago based on a micromagnetic model[4]. After the experimental discovery of magnetic skyrmions[5–8], it has been proposed to use such individual skyrmions in novel storage and logic devices[9–12]. This triggered further theoretical studies with a focus on isolated skyrmions confined in nanostructures[13,14], in ultrathin films[15], and in multilayers[16]. From an experimental point of view ultrathin transition-metal films[7,8] and transition-metal multilayers[17–21] have proven to be particularly useful to find novel skyrmion systems since the magnetic interactions can be tuned via interface composition and structure[8,18,20–23]. Typically, magnetic skyrmions arise in applied magnetic fields in materials that exhibit a zero-field spin spiral ground state[5,6,8]: at intermediate magnetic fields periodic skyrmion lattices arise before reaching the field-polarized magnetic state at increased magnetic field. At room temperature skyrmions with diameters between 30 and 400 nm[17–21] have been realized in multilayers, whereas at cryogenic temperatures isolated skyrmions with diameters down to a few nanometers have been observed in ultrathin epitaxial films[8,24,25].

The formation of magnetic skyrmions is governed by the interplay of exchange interaction, DMI, and anisotropy energy. Skyrmions at zero magnetic field, as desired for applications, can be metastable magnetic objects within a FM background only in a limited region of the phase space spanned by the magnetic interactions[4,26]. Experimental evidence of isolated zero-field skyrmions has been obtained recently in remanence: whereas in a ferrimagnetic material the skyrmion diameter was reported to go down to 16 nm[27], in FM multilayers the observed skyrmions had a diameter on the order of 100 nm[21]. Isolated zero-field skyrmions were also stabilized in ferromagnets by confinement[19] with diameters down to 50 nm[28], or in an effective magnetic field due to interlayer exchange coupling[29] where skyrmions with diameters of about 200 nm have been reported[30]. However, there are no reports on isolated zero-magnetic field skyrmions in the virgin state of a FM material, corresponding to the predicted metastable skyrmions.

Frustrated exchange interactions due to the competition of nearest neighbor exchange and exchange beyond nearest-neighbors—typical for itinerant magnets with long range exchange interactions[31]—can greatly enhance the energy barrier protecting skyrmions from collapse into the FM state[32] even allowing skyrmions in the limit of vanishing DMI[33,34]. At interfaces the exchange interactions can be tuned by the hybridization between $3d$- and $4d$-/$5d$-transition-metal layers[35]. For example, in both Rh/Fe and Pd/Fe bilayers on Ir(111) exchange interactions beyond nearest neighbors lead to exchange frustration. In Rh/Fe/Ir(111)[36] the exchange frustration is the driving force for a spin spiral ground state which is robust in an external magnetic field. In Pd/Fe/Ir(111), the DMI plays a key role allowing field-stabilized skyrmions[8,37] and the exchange frustration greatly increases skyrmion stability[32]. For Co, such exchange frustration has not yet been reported. However, the nearest-neighbor exchange interaction in Co monolayers can also strongly vary depending on the hybridization, for example it decreases by about 30% from Co/Pt(111)[38] to Co/Ir(111)[39] and by about 50% from Co/Pt(111) to Co/Ru(0001)[40], which demonstrates the role of the interface for the strength of exchange interactions.

Here we demonstrate that isolated magnetic skyrmions with a diameter of only 5 nm can be stabilized at zero magnetic field in FM Rh/Co atomic bilayers on the Ir(111) surface. Nanometer-size domain walls (DWs) with a unique rotational sense and individual skyrmions are observed at temperatures of 4 K using spin-polarized scanning tunneling microscopy (SP-STM)[41,42]. We

show via density functional theory (DFT) that due to hybridization at the Rh/Co/Ir interfaces the exchange interactions are strongly frustrated in these films. The DMI induces clockwise rotating spin structures and the large magnetocrystalline anisotropy prefers an out-of-plane magnetization. Atomistic spin dynamics simulations based on DFT parameters show that the frustrated exchange interaction stabilizes isolated zero-field skyrmions with a diameter of 5 nm in these films. The key role played by exchange frustration is an increase of the energy barrier for radial skyrmion collapse, which results in a chimera skyrmion annihilation mechanism at zero field.

## Results

**Scanning tunneling microscopy.** Figure 1 shows an STM measurement of a typical Rh/Co/Ir(111) sample in the magnetic virgin state. In this representation the constant-current topography is colourized with the spin-resolved differential conductance ($dI/dU$). The Co has a coverage of about half a monolayer and grows as stripes from the Ir step edges, dominantly pseudomorphic in fcc stacking[43]. The submonolayer amount of Rh also grows pseudomorphic both on the remaining bare Ir surface as well as on the Co stripes. On the Co it forms compact monolayer high islands which are either in hcp stacking (indicated by green arrows) or in fcc stacking (indicated by red arrows); the stacking difference of the Rh can be identified due to the variation of the density of states for the different adsorption sites (for the assignment of the stacking see Supplementary Fig. 1). On both the Co and the Rh/Co films we observe lateral variations in the topography and the differential conductance (see Supplementary Fig. 2), which we attribute to some intermixing of the two materials. A quantification of the number of exchanged atoms is not possible because we are not able to unambiguously identify the species of each atom at the surface, possibly due to their similar electronic properties as isoelectronic elements. On top of the Rh/Co there are a few small second layer Rh islands.

The magnetic state of the Co monolayer on Ir(111) is out-of-plane FM[43]. This measurement has been performed with a magnetic tip and due to the spin-polarized contribution to the tunnel current, i.e., the tunnel magnetoresistance (TMR), opposite magnetic domains manifest in the observed two-stage magnetic contrast (see white arrows in Fig. 1 indicating the derived magnetization direction of the Co stripes). Each of the Rh stackings also exhibits a two-stage contrast and in combination with experiments using an external magnetic field $B$ (see below and Supplementary Fig. 3) we conclude that the Rh/Co atomic bilayer behaves as one magnetic entity and is out-of-plane FM; the two-stage contrast represents Rh/Co domains oriented in the two opposite magnetization directions (see directions of the colored arrows in Fig. 1). We find that nearly all Rh$_{fcc}$/Co islands are in a single domain state. In contrast, several Rh$_{hcp}$/Co islands exhibit domains of both magnetization directions separated by DWs, see Fig. 2a for a closer view of the sample areas indicated by the black boxes in Fig. 1. These DWs surprisingly do not minimize their length but instead follow meandering paths. This suggests that the position of the DWs is dominated by a combination of small DW energy, i.e., only a small energy penalty for their existence, and an inhomogeneous potential landscape within the film due to the intermixing within the Rh/Co film.

Due to the canted tip magnetization (see sketch in the inset to Fig. 2a) not only the magnetic out-of-plane domains can be identified, but we also obtain magnetic contrast for the in-plane DWs. We find that the magnetic signal on the DWs is correlated with the local direction and environment of the wall: all brighter DWs are located at the right side of darker domains in Fig. 2a. Thus the magnetic structures in Rh$_{hcp}$/Co/Ir(111) have a fixed

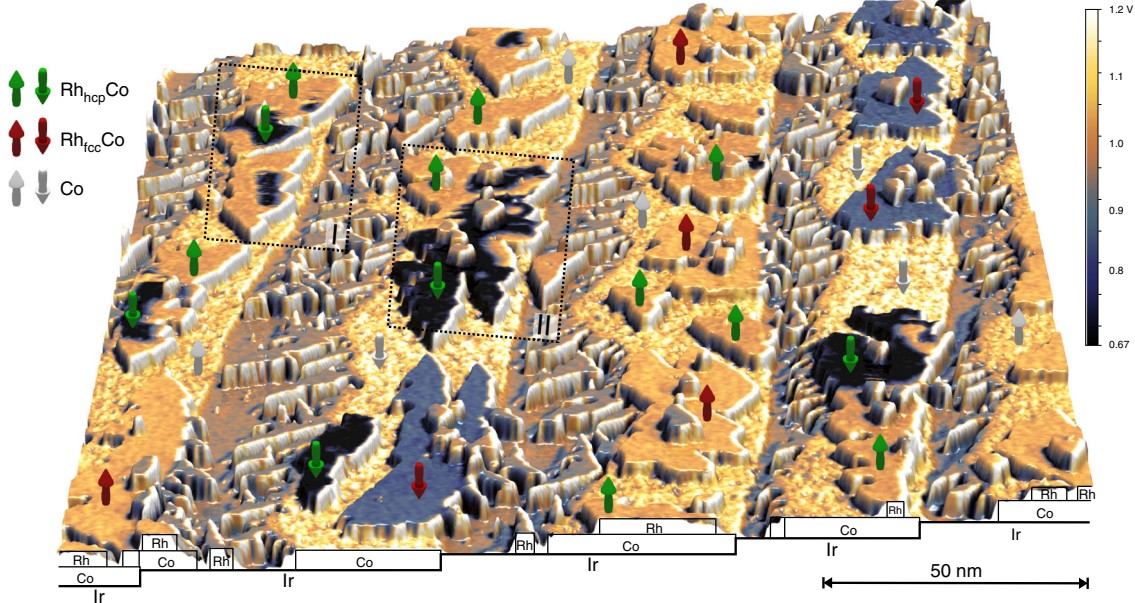

**Fig. 1** STM measurement of Rh/Co/Ir(111) in the magnetic virgin state. Perspective view of STM topography colourized with a simultaneously measured spin-resolved $dI/dU$ map of 0.4 atomic layers of Rh deposited on 0.5 atomic layers of Co on Ir(111). The front cross-section has been sketched and labeled for clarity. Co (indicated by white arrows) grows as stripes form the Ir(111) step edges. Rh covers some of the exposed Ir(111) surface. The stacking of the Rh on the Co monolayer is indicated by the color of the arrows. The directions of the arrows indicate the local magnetization direction. The two black rectangles mark the zoom-ins shown in Fig. 2. ($U = -250$ mV, $I = 800$ pA, $B = 0$ T, $T = 4.2$ K), Cr bulk tip, fast scan axis is horizontal)

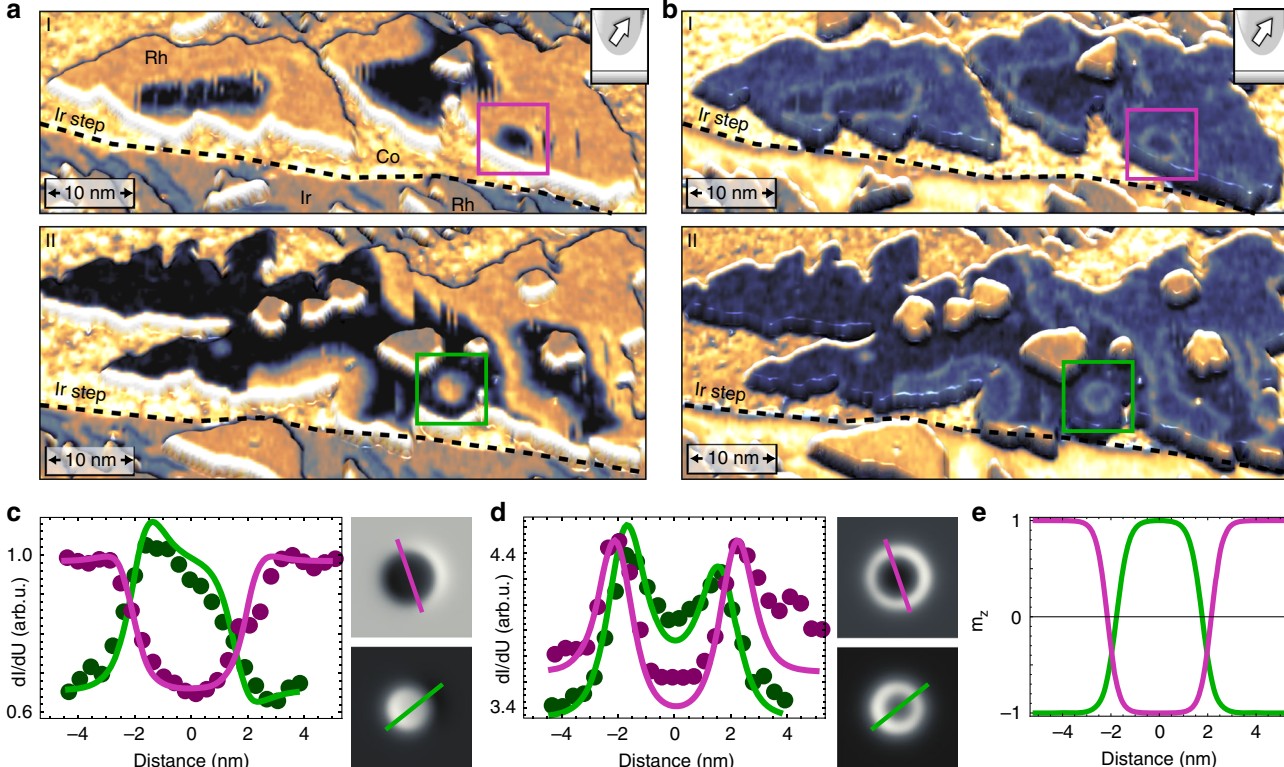

**Fig. 2** Zero field skyrmions in hcp-Rh/Co/Ir(111). **a**, **b** Perspective views of STM topography colourized with simultaneously measured spin-resolved $dI/dU$ maps at different bias voltages; these are the sample areas indicated by the rectangles in Fig. 1. Whereas in **a** the TMR contribution dominates and the FM domains can be identified by their two-stage contrast, in **b** the NCMR contribution is strong and the DWs appear as bright lines. The magnetic tip is identical for these measurements and it is sensitive to both the out-of-plane and an in-plane component of the sample's magnetization (see sketches in the insets). Isolated skyrmions with opposite magnetization are indicated by the boxes (**a** $U = -250$ mV; **b** $U = -400$ mV; **a**, **b** $I = 800$ pA, $B = 0$ T, $T = 4.2$ K, Cr bulk tip, fast scan axis is vertical, color gradient as in Fig. 1). **c**, **d** $dI/dU$ signal (dots) across the isolated skyrmions in **a**, **b** together with line profiles (solid lines) of STM simulations of two skyrmions (see Supplementary Note 1 and Supplementary Fig. 4). **e** Out-of-plane magnetization component $m_z$ across the two skyrmions used in the STM simulations

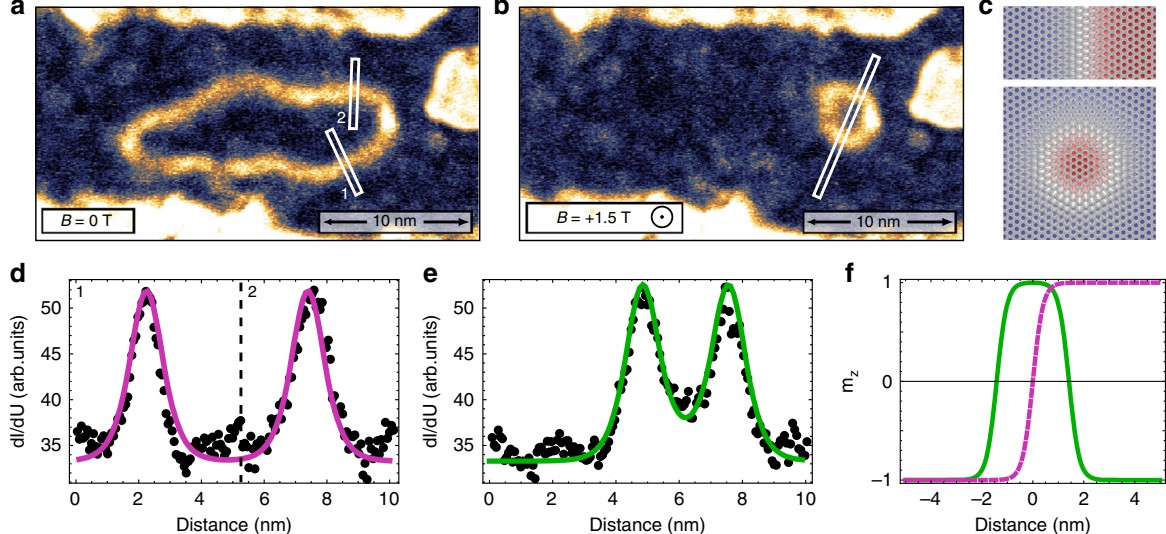

**Fig. 3** Domain wall width and skyrmion diameter. **a**, **b** $dI/dU$ maps of an isolated magnetic object in $Rh_{hcp}$/Co/Ir(111) at zero field and at $B = +1.5$ T measured with a non-spin-polarized tip; because of the NCMR contribution the DWs have a higher signal than the FM domains ($U = -250$ mV, $I = 800$ pA, $T = 4$ K, Cr bulk tip, color gradient as in Fig. 1). **c** sketches of a DW and a skyrmion ($w = 0.8$ nm), the color of the atomic magnetic moments indicates the out-of-plane magnetization component. **d**, **e** $dI/dU$ signal (dots) across the two DWs in **a** and the skyrmion in **b** positions are indicated by the white rectangles; the solid lines are profiles of STM simulations of the DW and skyrmion shown in **c**. **f** Corresponding out-of-plane magnetization component $m_z$ across the DW (magenta) and the skyrmion (green) of **c**

sense of magnetization rotation. It originates from the DMI, which favors Néel-type DWs and skyrmions with unique rotational sense in thin film systems[8,39,44].

When measuring the same sample areas at a different bias voltage, see Fig. 2b, the $dI/dU$ signal on the DWs is dominated by the tunnel noncollinear magnetoresistance (NCMR) contribution. This contrast is not related to the TMR but arises from changes of the local electronic properties within the noncollinear spin texture[45,46]. For the measurement of Fig. 2b we find a higher NCMR-related $dI/dU$ signal for sample positions where adjacent spins have a larger canting, i.e., bright lines mark the positions of DWs. The strength of an NCMR contribution in first approximation scales with the local mean angle between adjacent magnetic moments[45]. We find a significant NCMR contribution in a large energy range around the Fermi-energy. This electronic effect can be exploited to detect DWs and skyrmions also with a nonmagnetic metallic electrode and is demonstrated here in a Co film, a widely used magnetic material.

Small circular domains are found in both of the two oppositely magnetized FM domains of $Rh_{hcp}$/Co/Ir(111), see boxes in Fig. 2a, b, and because of the experimental finding of unique rotational sense of the magnetization rotation due to the DMI we conclude that they represent magnetic skyrmions. Consequently, the experimental $dI/dU$ maps can be reproduced by STM simulations of skyrmions, see gray scale images in Fig. 2c, d, with different contributions of TMR and NCMR for the two different bias voltages and a canted tip magnetization (see Supplementary Note 1 and Supplementary Fig. 4 for details); for the bias voltage used to obtain the data presented in Fig. 2a the TMR dominates, whereas at the bias voltage used for Fig. 2b the NCMR contribution to the signal is stronger. The skyrmions are modeled by circular domain walls and their diameters are 4.3 nm and 3.5 nm, and the comparison between experimental and simulated line profiles is reasonable (Fig. 2c, d). In Fig. 2e the derived out-of-plane magnetization components are plotted, which show the typical continuous magnetization rotation across a skyrmion. It is quite remarkable, that these opposite magnetic skyrmions coexist in the virgin state of

our Rh/Co film and that they do not collapse regardless of their small diameter.

Beside the occurrence of isolated skyrmions in the FM virgin state of $Rh_{hcp}$/Co (see also Supplementary Figs. 3 and 5), they can be obtained by shrinking larger FM domains in opposite magnetic field as seen in the measurements of Fig. 3a, b: the closed loop domain wall imaged bright in the $dI/dU$ map of Fig. 3a due to the NCMR encloses an isolated FM domain in zero field, which shrinks in size upon application of an out-of-plane magnetic field (Fig. 3b); note that the very bright signal at the right side of the skyrmion presumably indicates a pinning site for the DW; for a more straight-forward interpretation of the data we have used a tip with a negligible TMR contribution. The $dI/dU$ signal within the white rectangles is plotted versus the lateral position across the magnetic objects, see black dots in Fig. 3d, e. The solid lines show the simulated NCMR signal for two straight 180° DWs (Fig. 3d) and a cut through the magnetic skyrmion (Fig. 3e) for a DW width $w$ of 0.8 nm; the corresponding spin structures are shown in Fig. 3c, where the atomic magnetic moments are colored according to their out-of-plane magnetization components. The experimental data in Fig. 3d, e is reproduced well by the simulated NCMR signal both for the DWs as well as for the skyrmion. Figure 3f displays the out-of-plane magnetization components for the DW (magenta) and the skyrmion (green), and the derived skyrmion diameter is 2.8 nm, which corresponds to about ten atomic distances between opposite in-plane magnetizations. We would like to emphasize that there is no plateau in the skyrmion center, instead the spins rotate continuously.

**First-principles calculations.** To understand why in these Rh/Co films small magnetic skyrmions are stable at zero magnetic field, we apply density functional theory (DFT) (see "Methods"). Figure 4a shows the calculated energy dispersion $E(q)$ of homogeneous spin spirals in Rh/Co/Ir(111) along the high symmetry directions of the two-dimensional Brillouin zone (2D-BZ) neglecting spin–orbit coupling (SOC); to reveal the role of the Rh overlayer

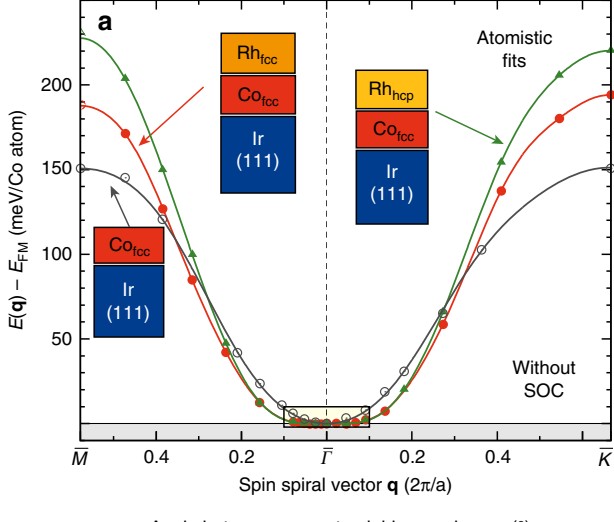

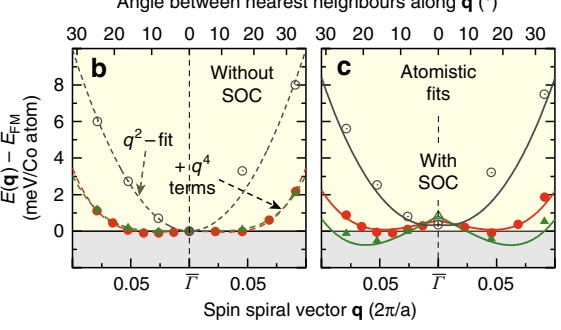

**Fig. 4** Energy dispersion of spin spirals for ultrathin Co films. **a** Energy dispersion of planar homogeneous spin spirals for Co/Ir(111), $Rh_{fcc}$/Co/Ir (111) and $Rh_{hcp}$/Co/Ir(111) along the high symmetry directions $\bar{M} - \bar{\Gamma} - \bar{K}$ of the two-dimensional Brillouin zone without spin-orbit coupling (SOC). Energies are given with respect to the FM state. The symbols (open gray circles for Co/Ir(111), red filled circles for $Rh_{fcc}$/Co/Ir(111) and green triangles for $Rh_{hcp}$/Co/Ir(111)) represent the DFT calculations while the solid lines are the fits to the Heisenberg exchange interaction beyond nearest neighbors (see Supplementary Note 2 for details). **b** Zoom around the FM state ($\bar{\Gamma}$-point) of **a**, where dashed lines represent a $q^2$ fit in the case of Co/Ir(111) and a fit including an additional $q^4$ term in the case of Rh/Co/Ir(111). **c** Zoom around the FM state ($\bar{\Gamma}$-point) of the energy dispersion for cycloidal spin spirals including the effect of spin-orbit coupling. The DMI leads to the local energy minima for clockwise rotating spin spirals along both high symmetry directions and the magnetocrystalline anisotropy energy is responsible for the constant energy shift of the spin spirals with respect to the FM state

the system of uncovered Co/Ir(111) is also shown. All Co films have a large FM nearest neighbor exchange constant, as evident from the large energy difference of the FM state at $\bar{\Gamma}$ and the antiferromagnetic states at the BZ boundary. At small $|\mathbf{q}|$, i.e., for small angles between nearest neighbor moments, Co/Ir(111) shows a rise of the energy with $q^2$, as expected for a typical ferromagnet (Fig. 4b). In contrast, $E(\mathbf{q})$ is extremely flat for Rh/Co/Ir(111) for both Rh stackings and to describe the dispersion, a $q^4$ term is required (Fig. 4b). Such an energy dispersion is characteristic for strong exchange frustration, where antiferromagnetic interactions beyond nearest neighbors compete with FM exchange between nearest neighbors (see Supplementary Tables 1 and 2 for values). This is quite unexpected for Co films, but, as we will show, it turns out to be beneficial for the stabilization of small magnetic skyrmions in zero field.

Calculations including SOC show a strong out-of-plane magnetocrystalline anisotropy energy for both Rh/Co stackings

(with 1.2 and 1.6 meV per Co atom, for fcc- and hcp-Rh stacking, respectively). In addition, they reveal an energy contribution to cycloidal spin spirals due to the DMI (Fig. 4c) favouring a clockwise rotational sense (see also Supplementary Fig. 6).

**Atomistic spin dynamics.** To explore the resulting spin structures, we apply atomistic spin dynamics[47] using the parameters from DFT (see Methods and Supplementary Tables 1 and 2). For $Rh_{fcc}$/Co we find out-of-plane FM domains and clockwise rotating DWs with a width of 1.4 nm. The DWs are exceptionally thin because of the large magnetocrystalline anisotropy in combination with the flat spin spiral dispersion for $|\mathbf{q}| < 0.2 \frac{2\pi}{a}$. The latter is responsible for an almost vanishing energy cost for spin cantings between nearest neighbors of up to almost 20° (cf. Fig. 4c). The DW energy obtained from our spin dynamics simulations with respect to the FM state amounts to only 2.0 meV nm$^{-1}$, one order of magnitude smaller than for Co/Ir(111) or Pt/Co/Ir(111)[39]. For $Rh_{hcp}$/Co the DW energy is negative due to a very small spin spiral energy minimum of $E = -0.7$ meV per Co atom with respect to the FM state (cf. Fig. 4c).

This is in contrast to the experimental observation of a FM state not only in $Rh_{fcc}$/Co but also in $Rh_{hcp}$/Co. There are several possible reasons for this discrepancy. Since there is intermixing at the Rh/Co interface the DMI, which stems mainly from the Co/Ir interface, will be reduced. For example for Co/Pt interfaces a reduction of 20% has been found[48] already for an intermixing of only 10–20%. Intermixing will also lower the exchange frustration thereby supporting the FM state (cf. Supplementary Note 2 and Supplementary Fig. 7). The DMI in our calculations may also be overestimated due to the use of first-order perturbation theory, an effect which has been quantified to about 10–25% depending on the system[38,49]. To obtain a FM ground state in the simulations for hcp-Rh/Co/Ir(111), as in the experiment, we have reduced the calculated DMI by 50%, i.e., just below the critical value where a transition from a spin spiral ground state to a FM state occurs (see Supplementary Table 2 for values). The resulting DW width in the simulations is 1.3 nm, see Fig. 5a, similar to the experimental value. With this reduced DMI the DW energy is positive, and its small value of only 4.4 meV nm$^{-1}$ can explain the experimental observation of meandering domain walls with a path dominated by the inhomogeneous potential landscape due to intermixing.

In agreement with the experimental findings our spin dynamics simulations show small zero magnetic field skyrmions within the FM ground state of the Rh/Co films (Fig. 5a). The small skyrmion diameter of about 4 nm is the result of the combination of flat energy dispersion of spin spirals close to the FM state (Fig. 4b) which reduces the energy cost of a fast spin rotation, and large magnetocrystalline anisotropy, which enforces the fast spin rotation. Note that all previously found nanometer-sized isolated skyrmions in ultrathin films[8,24,25,40] were induced by a magnetic field out of a spin spiral ground state while in the Rh/Co films individual skyrmions exist in the virgin FM state at zero magnetic field.

To gain further insight into the mechanism that stabilizes these small skyrmions at zero field, we calculate the energy barrier using minimum energy path calculations (see "Methods"). Typically, skyrmion annihilation occurs via a mechanism in which the skyrmion shrinks in size up to the saddle point (SP) and then collapses into the FM state[32]. For Rh/Co bilayers we find this radial symmetric collapse mechanism at magnetic fields above 1T (Fig. 5b, f). The energy along the corresponding minimum energy path starting from the skyrmion (Sk) crossing the SP, which defines the energy barrier, and ending in the FM state is shown in Fig. 5d with respect to the skyrmion energy. As

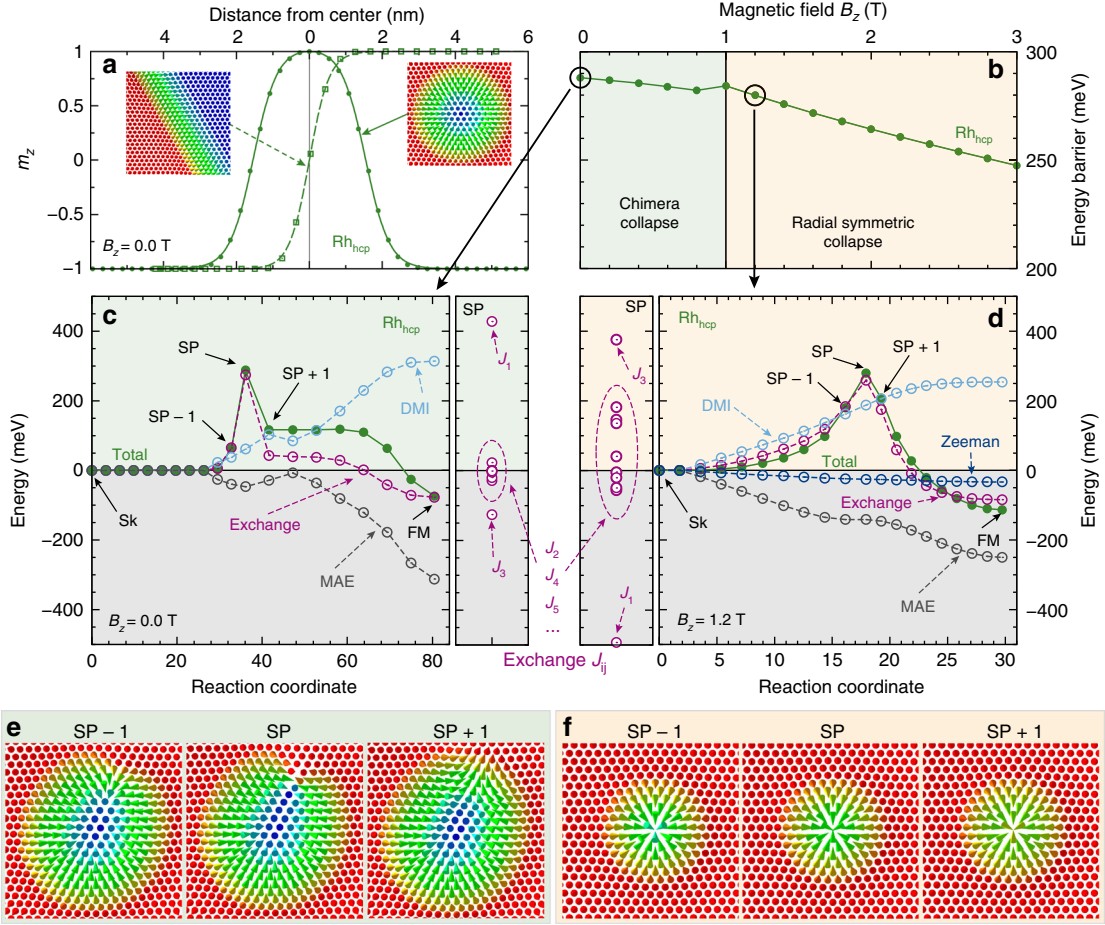

**Fig. 5** Profile and stability of skyrmions in Rh/Co/Ir(111). **a** Domain wall profile (open symbols) and zero field skyrmion profile (filled symbols), i.e., $z$-component $m_z$ of the local magnetic moment, obtained based on atomistic spin dynamics for $Rh_{hcp}$/Co/Ir(111). The dashed and solid lines are fits to the standard domain wall and skyrmion profile[57], respectively. Note that there are small deviations in both cases due to the exchange frustration. The insets show the domain wall and skyrmion spin structures on the two-dimensional atomic lattice. **b** Energy barrier protecting skyrmions in $Rh_{hcp}$/Co/Ir(111) against annihilation into the FM state as a function of applied magnetic field. The cross-over from the chimera collapse (green background) to the radial symmetric collapse (beige background) at 1 T is indicated. **c** Total energy and energy contributions for the chimera collapse from the different interactions (exchange, DMI, magneto-crystalline anisotropy energy (MAE)) at zero field versus the reaction coordinate along the minimum energy path from the initial isolated skyrmion (Sk) state to the final FM state for $Rh_{hcp}$/Co/Ir(111). Energies are summed over all atoms of the simulation box and are given relative to the energy of the isolated skyrmion state. The saddle point (SP) is indicated. To the right: exchange energy at the saddle point, $E_{SP}$, resolved with respect to the exchange interactions of different shells: $J_{1..10}$. **d** the same plot as in **c** for the radial symmetric collapse mechanism at a field of $B = 1.2$ T. **e** Images along the minimum energy path in **c** right before the SP (SP-1), at the SP, and just after the SP (SP+1). For all images of the minimum energy path see Supplementary Fig. 8. **f** Images along the minimum energy path in **d** for the radial symmetric collapse mechanism. Note that in **e, f** only a small part of the full simulation box which contained (70 × 70) spins is shown

expected, the Zeeman term and the magnetocrystalline anisotropy favor the transition into the FM state. The DMI energy amounts to about 200 meV at the SP. Since FM nearest-neighbor exchange ($J_1$) lowers the energy barrier for the radial symmetric collapse, it is surprising that the exchange interaction leads to an even larger contribution to the energy barrier. This effect is due to strong exchange frustration in Rh/Co/Ir(111) as seen from the decomposition of the total exchange energy at the SP in terms of different shells (left panel of Fig. 5d). Clearly, the contribution of $J_1$ is overcompensated by exchange beyond nearest neighbors.

To lower the energy barrier, a different annihilation mechanism is preferred at zero magnetic field and there is a transition from one to the other mechanism at 1T (Fig. 5b). Figure 5c shows how the energy evolves along the minimum energy path of this chimera collapse (see Fig. 5e). At the SP we obtain a large energy barrier of about 300 meV. The contribution due to exchange is more than three times larger than that of the DMI. Decomposing the exchange contribution by the neighboring shells (see Fig. 5c

right) shows that the barrier unexpectedly results from the large FM nearest-neighbor exchange interaction ($J_1$).

One can understand the individual contributions to the energy barrier by looking at the annihilation mechanism (Fig. 5e): just before the SP, at SP−1, the spin structure is a slightly oval shaped skyrmion with a similar size as the initial skyrmion and much larger than at the SP of the radial symmetric collapse mechanism (cf. Fig. 5f). At the SP a singular point is formed in the in-plane magnetized region of the skyrmion. This is energetically very unfavorable in terms of nearest-neighbor FM exchange interaction $J_1$. In contrast, the energy cost due to DMI is much lower since only a small part of the spins do not rotate with the preferred sense. The formation of the singular point transforms the skyrmion into a so-called chimera skyrmion[50] (see SP+1) with a vanishing topological charge which easily collapses into the FM state. This annihilation mechanism is preferred over the radial symmetric skyrmion collapse at zero field because it greatly reduces the DMI energy barrier and the high stability of zero-field

skyrmions in this system is a result of the frustrated exchange interaction.

## Discussion

Our work demonstrates that isolated magnetic skyrmions with a diameter of below 5 nm can be stabilized without applied magnetic field in ultrathin ferromagnetic Co films due to strong exchange frustration together with moderate DMI and large magnetocrystalline anisotropy. The measurements have been performed at temperatures of about 4 K, however, the large energy barriers found in our atomistic spin dynamics simulations based on DFT parameters suggest that isolated skyrmions in this system will be stable at significantly higher temperatures.

We anticipate that zero-field skyrmions stabilized by exchange frustration can also be obtained in multilayers composed of repeated Rh/Co/Ir sandwich structures. A similar transfer of skyrmion properties from ultrathin films to multilayers has been shown for Pd/Fe/Ir(111) based on DFT[22] and tuning magnetic skyrmion properties in Pt/Co/Fe/Ir multilayers at room temperature was experimentally demonstrated[20]. By tailoring interlayer exchange interactions in multilayers zero-field sub-10 nm skyrmions may become possible even at room temperature.

## Methods

**Sample preparation**. The Ir(111) single crystal surface was cleaned by cycles of annealing in oxygen with partial pressures in the range of $10^{-7}$ mbar up to about 1800 K to remove C impurities. For each sample preparation the surface was sputtered with Ar ions of about 800 eV with subsequent annealing to around 1500 K for 60 s. The Co was deposited onto the substrate at elevated temperatures to achieve step flow growth. The Rh was deposited after the sample had reached room temperature. Typical deposition rates for Co and Rh are between 0.1 and 0.2 atomic layers per minute. Samples were transferred in vacuo to a low temperature scanning tunneling microscope equipped with a Cr bulk tip for spin-resolved measurements. The Cr bulk tip was introduced into ultra-high vacuum after etching and in situ cleaning was performed by field emission.

**Density functional theory**. We apply DFT based on the full potential linearized augmented plane wave method as implemented in the FLEUR code (www.flapw.de). This all-electron method ranks among the most accurate implementations of DFT. Computational details for Co/Ir(111) can be found in Ref. [39]. For Rh/Co/Ir(111) we performed structural relaxations within the FM state using a symmetric film consisting of a Rh/Co bilayer on both sides of five layers of Ir(111) with the theoretical lattice constant ($a = 3.82$Å)[37]. Relaxed interlayer distances are given in Supplementary Table 3. The muffin tin (MT) radii were $R_{MT} = 2.31$ a.u. for Ir and Rh and $R_{MT} = 2.23$ a.u. for Co. The cutoff for the basis functions was $k_{max} = 4.0$ a.u.$^{-1}$. We used 240 **K**-points in the irreducible wedge of the two-dimensional Brillouin zone (2D-BZ) and the generalized gradient approximation[51].

For spin spiral calculations[52,53] with and without spin–orbit coupling and to obtain the magnetocrystalline anisotropy energy, an asymmetric slab with a Rh/Co bilayer on nine layers of the Ir(111) substrate was used. The energy dispersion of flat spin spirals (see Supplementary Note 2 for details) are calculated with a dense **k**-point mesh of $44 \times 44$ **k**-points in the full 2D-BZ and a basis cutoff of $k_{max} = 4.0$ a.u.$^{-1}$ was used. Close to the $\bar{\Gamma}$-point ($|\mathbf{q}| \to 0$), we checked the convergence of the energy dispersions with up to $100 \times 100$ **k**-points and $k_{max} = 4.3$ a.u.$^{-1}$. These calculations were performed in local density approximation[54].

**Spin-dynamics simulations**. In order to relax the spin structures of the domain walls and the isolated skyrmions and to calculate their energy differences with respect to the FM state, we used the Landau–Lifshitz equation:

$$\hbar \frac{d\mathbf{m}_i}{dt} = \frac{\partial \mathcal{H}}{\partial \mathbf{m}_i} \times \mathbf{m}_i - \alpha \left( \frac{\partial \mathcal{H}}{\partial \mathbf{m}_i} \times \mathbf{m}_i \right) \times \mathbf{m}_i \quad (1)$$

where $\mathbf{m}_i = \frac{\mathbf{M}_i}{M_i}$ is the unit vector of the magnetic moment at atom site $i$, $\alpha$ is the damping parameter and $\mathcal{H}$ is the Hamiltonian:

$$\mathcal{H} = -\sum_{ij} J_{ij} \left( \mathbf{m}_i \cdot \mathbf{m}_j \right) - \sum_{ij} \mathbf{D}_{ij} \left( \mathbf{m}_i \times \mathbf{m}_j \right) + K \sum_i \left( m_i^z \right)^2 \quad (2)$$

Here $J_{ij}$ denotes the strength of the exchange interaction between spins on atom sites $i$ and $j$ and $\mathbf{D}_{ij}$ is the vector characterizing their DMI. $K$ represents the strength of the uniaxial anisotropy. For the simulations presented in Fig. 5, we have used various damping parameters of $\alpha \in [0.05, 1]$ and a time step of 0.1 fs. The simulations are carried out with $1.5$–$5.0 \times 10^6$ timesteps and the equation of motion was solved with the semi-implicit integrator as proposed by Mentink et al.[55]. The

values of the exchange constants, the DMI, and the magnetocrystalline anisotropy energy are given in the Supplementary Tables 1 and 2 in units of meV per atom. We used a hexagonal lattice of $70 \times 70$ spins and a magnetic moment of 2.5 $\mu_B$ which corresponds to a combined value of Rh (0.6 $\mu_B$), Co (1.8 $\mu_B$), and the Ir interface layer (0.1 $\mu_B$) in the FM state.

**Geodesic nudged elastic band method**. Using the relaxed structures of skyrmions from spin-dynamics simulations, we calculate the minimum energy paths (MEPs) for annihilation processes using the geodesic nudged elastic band (GNEB) method[56]. Starting from a local energy minimum (skyrmion), a path is generated into the global energy minimum (FM state). The path is systematically brought into the MEP, while it is divided into a discrete chain of states, the so-called images. The first image corresponds to the skyrmion and the last image to the FM state. After the relaxation of the starting point, the effective field is calculated along a local tangent to the path at each image. Its components are substituted by an artificial spring force between the images to ensure a uniform distribution of the path. Once, the whole chain of images is converged, the path represents the MEP and the SP is the energy maximum. To determine the height and the position of the SP correctly, the climbing image technique is applied.

## Data availability

The datasets generated and analyzed during the current study are available from the authors upon reasonable request.

## Code availability

The atomistic spin dynamics code is available from the authors upon reasonable request.

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

## Acknowledgements

This project has received funding from the European Union's Horizon 2020 research and innovation program under grant agreement No 665095 (FET-Open project MAGicSky). K.v.B. and A.K acknowledge financial support from the Deutsche Forschungsgemeinschaft (DFG, German Research Foundation)—402843438; 408119516. S.M., S.v.M., and S.H. thank the Norddeutscher Verbund für Hoch- und Höchleistungsrechnen (HLRN) for providing computational resources. We thank Pavel F. Bessarab for valuable discussions.

## Author contributions

S.H. devised the project. M.P. performed the measurements. M.P., K.v.b. and A.K. analyzed the experimental data. S.M. performed the DFT calculations. S.M. and S.v.M. performed the spin dynamics and GNEB calculations. S.M., S.v.M and S.H. analyzed the calculations. M.P. and S.M. prepared the figures. M.P., K.v.b., S.M. and S.H. wrote the manuscript. S.M., M.P., S.v.M., A.K., R.W., K.v.b. and S.H. discussed the results and contributed to the manuscript.

## Additional information

**Competing interests:** The authors declare no competing interests.

