## [Peer Review File · Nature Communications]

Reviewers' comments:

Reviewer #1 (Remarks to the Author):

I recommend the authors to revise their manuscript to solve several concerns I described in a separate file.

Review of " Isolated zero field sub-10 nm skyrmions in ultrathin Co films" by Sebastian Meyer et al. submitted to Nature Communications. Manuscript ID: NCOMMS-19-08562-T Isolated skyrmions with diameters below 10 nm that are stable at zero magnetic field and room temperature are desired for their applications for future spintronic devices. The manuscript describes the experimental observation of isolated skyrmions with diameters smaller than 5 nm at zero magnetic field in Rh/Co atomic bilayers on Ir(111) surface. They used spin-polarized differential conductance with two different bias voltages, one for tunnel magnetoresistance (TMR) signal and another for non-collinear magnetoresistance (NCMR). They presented two experimental evidences for the zerofield skyrmions. One is two pairs of STM images in two different field-of-views (Fig. 1). Another figure (Fig.2) demonstrates the change of NCMR contrast STM images of an identical field-of-view (but different from the one as shown in Fig.1) when they changed the perpendicular magnetic field from 0 mT to 1.5 T. Then they tried to explain the stable skyrmion by using DFT calculations and spin-dynamic calculations, finally concluded that such stabilization was due to strong exchange frustration and large magnetocrystalline anisotropy. Their manuscript is properly documented with sufficient references. In addition, their experimental/theoretical techniques are both robust and relevant. The subject - "isolated zero field sub-10nm skyrmion" is novel and significant if it proves to be correct. However, as for scientific rigor, I have several concerns to be solved to be positive for publication in Nature Communications.

Major concerns

(1) Page 2, line 27: "Small circular domains are found in both of the two oppositely magnetized FM domains of Rhhcp/Co/Ir(111), see boxes in Fig. 1a and b." Frankly speaking, the boxed domains hardly appear as circular. Rather, they look like polygonal shapes similar to neighboring islands. From these images, it is more natural to conclude that these domains are not skyrmion but pinned domain wall at polygonal grain boundaries. Observing Figure 2 further reinforces this concern. The domain wall at the right-edge remained same (pinned at the same position) after application of 1.5 T. This strongly suggests that the polygonal domain is just an accidentally pinned domain wall at the periphery of small grains with a size below 10 nm. Therefore, in my opinion, Figures 1&2 cannot be considered as the clear evidences of "stable skyrmions at zerofield with diameters below 10 nm". Supplementary Figure S4 also shows a polygonal domain. I recommend the author to perform structural characterizations of the thin film to confirm that no structural boundary corresponding to the size of the observed "zerofield skyrmion" is present in their film.

(2) Figures 1&2: It is confusing that they mixed images acquired with different conditions. In figure 1, it seems that they used spin-polarized tip with different biasvoltages, while in figure 2, they used a non-spin-polarized tip to obtain NCMR contrast. To support their conclusions, the authors had better display the most evident images acquired with identical experimental conditions if there is no special reason to mix different kinds of images. I recommend the authors to use conventional spin-polarized STM images

for Figures 1&2 to display zero-field skyrmion clearly. Then, using a new Figure 3, they can demonstrate that NCMR detection of such skyrmion is also possible, which should be significant for future applications.

(3) Page 3, line 3: "The experimental data is reproduced well by the simulated NCMR signal and the derived skyrmion diameter is 2.8 nm, see Fig. 2f, which corresponds to about 10 atomic distances between opposite in-plane magnetisations. We would like to emphasise that there is no plateau in the centre of this circular domain, instead the spins rotate continuously." These sentences are hard to understand. The simulated profile (green line) of skyrmion as shown in Fig. 2f is different from those shown in Fig. 2d&e.

(4) Page 4, line 29: "We have shown that the detection of individual sub-10 nm skyrmions via the NCMR effect is possible also in Co films, meaning that these skyrmions can be directly detected in an all-electrical read-out." NCMR detection of zero-field skyrmion may be important for all-electrical read-out. However, the main point of the present manuscript is the observation of zero-field skyrmion.

Minor concerns

(1) Figure 1ab (also Figure 2): No data bar. Indicate what dark-blue/dark-yellow/white colors mean. In addition, label the upper and lower field-of-view images separately, and explain the differences, if any.

(2) It is misleading to write "To become technologically competitive, isolated skyrmions with diameters below 10 nm that are stable at zero magnetic field and room temperature are desired." in the 2nd sentence of abstract, while their experiments were done at liquid He temperature. It is desirable to discuss how the present study can lead to room temperature zero-field skyrmion at the end of the main text if they truly "anticipate that multilayers can be tailored to transfer these advantageous properties to structures suitable for applications at room temperature."

(3) Figure 2 legend: "f: Corresponding out-of-plane magnetisation component m_z across the domain wall and skyrmion of c-e." This should be corrected as "f: Corresponding out-of-plane magnetization component m_z across the domain wall (magenta) and skyrmion (blue) as shown in c."

(4) Page 3, line 3: "The experimental data is reproduced well by the simulated NCMR signal and the derived skyrmion diameter is 2.8 nm, see Fig. 2f, which corresponds to about 10 atomic distances between opposite in-plane magnetisations. We would like to emphasise that there is no plateau in the centre of this circular domain, instead the spins rotate continuously." Some typographic errors. They should be corrected as "magnetizations" and "emphasize"

Reviewer #2 (Remarks to the Author):

In this paper, Meyer et al. report on small skyrmions with sub-10 nm scale in ultrathin films of Co in zero magnetic field by scanning tunneling microscopy measurements. They also present theoretical calculation to elucidate the microscopic mechanism of the stabilization of small size skyrmions. The results are interesting, and I basically recommend publication of this article in Nature Communications after the following issues are appropriately addressed.

1) Generally, explanation in main text is not enough while plenty of data are presented in supplementary materials. More information should be presented and discussed in the main text, instead of the supplementary materials. Also, introduction part should be made more detailed and

comprehensive for general audience before presenting the results.

2) Actual domain structure corresponding to Fig. 1a, b is not clear to me. Authors mention "two-stage contrast", and I guess that these are black area and yellowish (or orange colored) area. But there are white and black lines also, and these may be domain walls? But if these are domain walls, then they should locate between two domains with different magnetization direction, namely between black and yellow areas. This is obviously not the case. A schematic figure of domains and domain walls with possible magnetization direction which corresponds to the dI/dV map will help understand the beautiful data more easily and correctly.

3) I guess the difference between upper panel and lower one is the spatial position of the dI/dV maps. Is this correct? This point is not clearly mentioned in either text or caption. Initially, I thought that these correspond to dI/dV maps taken with different condition for a single position.

4) I guess the magnetization direction of Cr tip is perpendicular to the film plane. Is this correct? Then, why do the domain walls show brighter contrast than the domain itself?

5) DFT calculation indicates that the existence of Rh layer changes the q -dependence of $E(q)$ qualitatively from q^2 to q^4 , but why does this occur? The result is fitted with atomistic spin model, but the original system is a metallic one. How is this understood?

Reviewer #3 (Remarks to the Author):

Report on paper: # NCOMMS-19-08562-T by S. Meyer et al.

This study entitled "Isolated zero field sub-10 nm skyrmions in ultrathin Co films" by S. Meyer et al is part of the large research effort currently made in solid state physics on the investigation of magnetic skyrmions both for addressing fundamental questions but also for the potential use of these topologically protected magnetic textures in novel spintronic conceptual devices. These developments require to find solutions to stabilize extremely small isolated skyrmions. One of the important results of this study is precisely to have found such a case in the thin film system Co/Rh on Ir (111). Indeed, combining advanced magnetic imaging by STM and DFT calculations, the authors bring clear evidences of the impact of exchange frustration for the stabilization of 5-nm diameter isolated skyrmions. Moreover, the balance between all the magnetic interactions in this system makes that these ultra-small skyrmions can be stable even at zero applied field. Even if this observation has been done at low temperature, this result provides a clear strategy to achieve similar results at room temperature in multilayer systems. For this reason, I believe that this work is of interest for the growing community of researchers working on skyrmions and more generally nanomagnetism. In conclusion, I recommend this work for publication in Nature Communications after having clarified a few points that are listed hereafter.

1) The most important point concerns the comparison between experiments and DFT calculations made in page 3. As far as I understand, it comes out from the calculations that for Rh hcp on Co, the ground state is not the one observed by STM. Then the authors comment this discrepancy only by saying that the DMI is typically overestimated using their approach! It would be important that they bring some further explanations. How much do they have to reduce DMI in order to "agree" with the experimental observation? In the present version, it is not clear whether the DMI overestimation is specific to the Rh/Co system or if it is more general result.

2) In the same paragraph, the authors propose that additional reduction of the actual DMI value can come from intermixing. Did they try to include some intermixing in the DFT calculations and if yes, how large is the intermixing to be introduced in order to agree with experiments?

3) As the possible intermixing between Rh and Co is also invoked to explain the shape of the DWs, I wonder if the STM experiments cannot be used to extract some more quantitative information on the actual intermixing?

4) It is not clear to me what are the reasons to show two different regions in Fig1a and 1b and why it brings more information than showing only one?

In Fig 1c, it is rather clear that the skyrmion profile is not symmetric. How such a non-symmetric skyrmion can be obtained in the STM simulations (green curve in Fig 1c)? Finally, in Fig. 1e, the statement made that the fitted profile shows the typical continuous rotation is not really justified, at least for the violet curve.

5) For a non-specialist reader, it will be important to remind in the text the expression of the DW energy as it is discussed several times.

6) Even if the authors mention that they focus on Rhhcp/Co on Ir(111), I would find interesting to discuss a bit more the other stacking Rhfcc/Co. For example, it is not clear if any STM experiments have been obtained on this stacking. If yes, it would be interesting to know if the predicted opposite rotation is observed?

7) Table S12 : how do you understand the large frustration on DMI as D_3 is much larger than D_1 for example?

Reply to Reviewer #1:

Isolated skyrmions with diameters below 10 nm that are stable at zero magnetic field and room temperature are desired for their applications for future spintronic devices. The manuscript describes the experimental observation of isolated skyrmions with diameters smaller than 5 nm at zero magnetic field in Rh/Co atomic bilayers on Ir(111) surface. They used spin-polarized differential conductance with two different bias voltages, one for tunnel magnetoresistance (TMR) signal and another for non-collinear magnetoresistance (NCMR). They presented two experimental evidences for the zero-field skyrmions. One is two pairs of STM images in two different field-of-views (Fig. 1). Another figure (Fig.2) demonstrates the change of NCMR contrast STM images of an identical field-of-view (but different from the one as shown in Fig.1) when they changed the perpendicular magnetic field from 0 mT to 1.5 T. Then they tried to explain the stable skyrmion by using DFT calculations and spin-dynamic calculations, finally concluded that such stabilization was due to strong exchange frustration and large magnetocrystalline anisotropy.

Their manuscript is properly documented with sufficient references. In addition, their experimental/theoretical techniques are both robust and relevant. The subject - "isolated zero field sub-10nm skyrmion" is novel and significant if it proves to be correct. However, as for scientific rigor, I have several concerns to be solved to be positive for publication in Nature Communications.

We thank the referee for carefully reading our manuscript, the valuable comments based on which we improved our manuscript, and the positive assessment of the relevance of the subject of our work. Below we address all issues raised by the referee point-by-point.

Major concerns

(1) Page 2, line 27: "Small circular domains are found in both of the two oppositely magnetized FM domains of Rhhcp/Co/Ir(111), see boxes in Fig. 1a and b."

Frankly speaking, the boxed domains hardly appear as circular. Rather, they look like polygonal shapes similar to neighboring islands. From these images, it is more natural to conclude that these domains are not skyrmion but pinned domain wall at polygonal grain boundaries. Observing Figure 2 further reinforces this concern. The domain wall at the right-edge remained same (pinned at the same position) after application of 1.5 T. This strongly suggests that the polygonal domain is just an accidentally pinned domain wall at the periphery of small grains with a size below 10 nm. Therefore, in my opinion, Figures 1&2 cannot be considered as the clear evidences of "stable skyrmions at diameters below 10 nm". Supplementary Figure S4 also shows a polygonal domain.

I recommend the author to perform structural characterizations of the thin film to confirm that no structural boundary corresponding to the size of the observed "zero-field skyrmion" is present in their film.

The reviewer is correct, that the sub-5 nm magnetic objects are not perfectly circular. In the following we first discuss the reason for this deviation from perfect axial symmetry and then argue why we are still convinced that these are magnetic skyrmions.

The RhCo bilayer does not exhibit a perfectly flat surface, instead we observe lateral variations of the apparent height in constant-current topographic images as well as variations of the differential conductance of the film. We attribute this to a non-perfect interface between the Rh and the Co atomic layers due to intermixing.

Regardless of the intermixing the bilayer is still pseudomorphic, which is evident from the clear distinction between fcc-stacked and hcp-stacked Rh on Co. The variation in the apparent height due to the intermixing is on the order of 7 pm. Neither from the constant-current topography images nor from the maps of differential conductance we are able to identify the different atomic species, possibly due to their similar electronic properties as isoelectronic elements.

While the atomic structure is periodic (neglecting the fact of Rh or Co atom substitution), this lateral variation of the electronic states due to intermixing is expected to result in an inhomogeneous potential landscape for the position of domain walls. This means, that the exact path of a domain wall will be influenced by the local electronic structure, which is in turn governed by the details of exact atomic nature in the surrounding. In other words, the intermixing leads to local pinning of the domain walls. This, together with a small domain wall energy, can give rise to the observed meandering domain wall paths, as discussed in the manuscript. The width of the walls varies slightly depending on position, but in this pseudomorphic film this is not due to grain boundaries but the local inhomogeneities; however, the variation is small enough to consider the measured width a very good estimate of the true wall width. The agreement with the calculations also supports this conclusion.

In order to make this more clear we have expanded the discussion about the intermixing and the consequences for the domain wall paths in the revised version of the manuscript (see page 2, last paragraph) and added a high-resolution topographic image in Suppl. Fig. S2 of the revised version.

Our measurements with a canted magnetic STM tip (Fig. 1 and Fig. 2 of the revised version) demonstrate that the spin texture has unique rotational sense. This means that the sub-5 nm magnetic objects not only have the topology of skyrmions, but also occur with unique rotational sense governed by the DMI. Therefore, even though there are slight deviations from axial symmetry and variations in the size and shape due to the reasons mentioned above, we are convinced that they represent magnetic skyrmions. We see that these sub-5 nm skyrmions are stable in the experiments on the timescale of our measurements (hours to days). For these reasons our claim of the observation of stable sub-5 nm skyrmions in zero magnetic fields is justified.

(2) Figures 1&2: It is confusing that they mixed images acquired with different conditions. In figure 1, it seems that they used spin-polarized tip with different biasvoltages, while in figure 2, they used a non-spin-polarized tip to obtain NCMR contrast. To support their conclusions, the authors had better display the most evident images acquired with identical experimental conditions if there is no special reason to mix different kinds of images. I recommend the authors to use conventional spin-polarized STM images for Figures 1&2 to display zero-field skyrmion clearly. Then, using a new Figure 3, they can demonstrate that NCMR detection of such skyrmion is also possible, which should be significant for future applications.

Based on several comments from the Reviewers we have rearranged the figures in the revised version: we have moved a larger version of previous Suppl. Fig. S2 to become Fig. 1 of the main text. This allows to explain the topography of this overview image and the spin-resolved differential conductance of the Rh/Co areas more clearly, in addition to the discussion about the Rh-stacking-dependence of the magnetic state.

Now current Fig. 2 (similar to previous Fig. 1) shows the zero-field magnetic skyrmions of two enlarged areas at two different bias voltages. For the system of the Rh/Co bilayer a clear separation of the spin-polarized (TMR) contribution to the signal from the NCMR contribution is not possible, because we find that the NCMR is rather strong at nearly all bias voltages, whereas the strength of the TMR contribution is large at some particular bias voltage (note that the simulation of the line profiles of present Fig. 2 also requires an NCMR contribution of about 20% of the TMR contribution, see Supplementary Text S5 and new Supplementary Fig. S6).

Thus the NCMR is present in all measurements, and in order to separate the different contributions we present data where the TMR vanishes and only the NCMR is present, current Fig. 3. We believe that the reader can more easily interpret images with only one magnetoresistance contribution, instead of data with mixed TMR and NCMR. For this reason we do not follow the recommendation of the Reviewer in this point, however, we have revised the corresponding text in the manuscript (page 3, last paragraph, and page 4, second paragraph) and added Supplementary Fig. S6 to make this more clear.

(3) Page 3, line 3: "The experimental data is reproduced well by the simulated NCMR signal and the derived skyrmion diameter is 2.8 nm, see Fig. 2f, which corresponds to about 10 atomic distances between opposite in-plane magnetisations. We would like to emphasise that there is no plateau in the centre of this circular domain, instead the spins rotate continuously." These sentences are hard to understand. The simulated profile (green line) of skyrmion as shown in Fig. 2f is different from those shown in Fig. 2d&e.

We thank the Reviewer for pointing this out and we have rephrased this part in the revised manuscript (page 4, second paragraph).

(4) Page 4, line 29: "We have shown that the detection of individual sub-10 nm skyrmions via the NCMR effect is possible also in Co films, meaning that these skyrmions can be directly detected in an all-electrical read-out." NCMR detection of zero-field skyrmion may be important for all-electrical read-out. However, the main point of the present manuscript is the observation of zero-field skyrmion.

The Reviewer is correct, that the main point of the present manuscript is the observation of zero-field skyrmions. We have therefore removed this sentence from the revised discussion paragraph (see also answer to minor concern (2) below).

Minor concerns

(1) Figure 1ab (also Figure 2): No data bar. Indicate what dark-blue/dark-yellow/white colors mean. In addition, label the upper and lower field-of-view images separately, and explain the differences, if any.

We thank the Reviewer for pointing this out and have added a color scale bar in Fig. 1. In Fig. 2 the scale is evident from the numbers on the vertical axes of (c) and (d).

(2) It is misleading to write "To become technologically competitive, isolated skyrmions with diameters below 10 nm that are stable at zero magnetic field and room temperature are

desired." in the 2nd sentence of abstract, while their experiments were done at liquid He temperature. It is desirable to discuss how the present study can lead to room temperature zero-field skyrmion at the end of the main text if they truly " anticipate that multilayers can be tailored to transfer these advantageous properties to structures suitable for applications at room temperature."

We thank the referee for pointing this out and agree that it is possible that readers might be misled if they just glance at our paper and do not read it in detail. In order to clearly state the difference between the general statement in the abstract which the referee quoted and the achievements in our work we have changed the following sentence in the abstract:

"Here we report zero field isolated skyrmions with diameters smaller than 5 nm **observed at temperatures of about 4 K** coexisting with 1 nm thin domain walls in Rh/Co atomic bilayers on the Ir(111) surface."

By this change there should be no confusion possible. We have also mentioned the temperature of our experiments in the discussion at the end of the paper (see below). We would like to note that while the experiments were performed at low temperatures it is *a priori* not clear up to which temperature isolated skyrmions are stable in this system. The large energy barriers found in our simulations based on DFT parameters suggest a much larger transition temperature. Therefore, temperature-dependent experiments are a promising future study.

The second point raised by the referee shows that we need to clarify our ideas towards room temperature zero-field skyrmions. Our idea is twofold. The first aspect that we think can be transferred from ultrathin films to multilayers is the strong exchange frustration in the Co films, which is beneficial for stabilization of sub-5nm zero-field skyrmions. Previously, we have demonstrated based on DFT calculations (B. Dupé *et al.*, Nature Comm. **7**, 11779 (2016), Ref. [23] in our manuscript) that the properties of the well-known skyrmion film system Pd/Fe/Ir(111) can be transferred to multilayers composed of Pd/Fe/Ir sandwich structures. We have also shown that the interactions can be tuned by the composition of the 4d-3d interface. A similar transfer of desired properties from Rh/Co/Ir(111) to Rh/Co/Ir multilayers should be possible.

The experimental study on Pt/Co/Fe/Ir based multilayers from the group of Panagopoulos (Nature Mat. **16**, 898 (2017), Ref. [13] in our manuscript) further shows that tailoring of magnetic interactions is possible in multilayers and that skyrmion properties can be tuned over a wide range. The skyrmions in that study were on the order of 50 nm and room temperature stability has been achieved.

The second aspect concerns the stability at room temperature. As the referee emphasizes this is a nontrivial problem. One route is to use the interlayer exchange coupling between different magnetic layers within multilayer structures. We have shown previously (B. Dupé *et al.*, Nature Comm. **7**, 11779 (2016)) that this allows to significantly enhance transition temperatures. Since it is not clear whether room temperature can be reached we have phrased this statement more carefully in our revised manuscript.

We have changed the discussion part at the end of our manuscript in order to clarify our ideas on transferring skyrmion properties from films to multilayers (see page 6).

(3) *Figure 2 legend: "f: Corresponding out-of-plane magnetisation component m_z across the domain wall and skyrmion of c-e." This should be corrected as "f: Corresponding out-of-plane magnetization component m_z across the domain wall (magenta) and skyrmion (blue) as shown in c."*

We have corrected this in the revised version.

(4) *Page 3, line 3: "The experimental data is reproduced well by the simulated NCMR signal and the derived skyrmion diameter is 2.8 nm, see Fig. 2f, which corresponds to about 10 atomic distances between opposite in-plane magnetisations. We would like to emphasize that there is no plateau in the centre of this circular domain, instead the spins rotate continuously." Some typographic errors. They should be corrected as "magnetizations" and "emphasize"*

We have corrected this in the revised version.

Reply to Reviewer #2:

In this paper, Meyer et al. report on small skyrmions with sub-10 nm scale in ultrathin films of Co in zero magnetic field by scanning tunneling microscopy measurements. They also present theoretical calculation to elucidate the microscopic mechanism of the stabilization of small size skyrmions. The results are interesting, and I basically recommend publication of this article in Nature Communications after the following issues are appropriately addressed.

We thank the referee for reviewing our paper and for recommending, in principle, publication of our work in Nature Communications. We have improved our manuscript based on the helpful comments and questions of the referee and hope that the referee agrees with us that the paper is now in a state appropriate for publication.

Below we address all issues raised by the referee point-by-point.

1) Generally, explanation in main text is not enough while plenty of data are presented in supplementary materials. More information should be presented and discussed in the main text, instead of the supplementary materials. Also, introduction part should be made more detailed and comprehensive for general audience before presenting the results.

We thank the referee for making this point. We feel that the revisions which we have made based on this comment, and the extended list of references, have significantly improved our paper and hope that the referee is of the same opinion.

We have moved a larger view of previous Suppl. Fig. 2 to the main text (present Fig. 1). In the theory part of the manuscript we have extended the Fig. 5 (previously Fig. 4) which now includes also the radial symmetric skyrmion collapse mechanism at finite magnetic field and a panel which displays the cross-over between the two collapse mechanisms. The discussion in the main text has been adapted accordingly. Thereby, the content of previous

Supplementary Fig. S9 is moved to the main paper and has been deleted from the revised Supplementary Information.

We have revised the manuscript according to the style of Nature Communications by introducing sections and extended the introduction considerably in order to explain in more detail the issues concerning the stabilization of zero-field magnetic skyrmions and the role of frustrated exchange interactions at transition-metal interfaces. We have also included a paragraph which summarizes the key achievements of this work.

2) Actual domain structure corresponding to Fig. 1a, b is not clear to me. Authors mention "two-stage contrast", and I guess that these are black area and yellowish (or orange colored) area. But there are white and black lines also, and these may be domain walls? But if these are domain walls, then they should locate between two domains with different magnetization direction, namely between black and yellow areas. This is obviously not the case. A schematic figure of domains and domain walls with possible magnetization direction which corresponds to the dI/dV map will help understand the beautiful data more easily and correctly.

We thank the Reviewer for pointing this out. In the revised version of the manuscript we have introduced a new Fig. 1, which in addition to the spin-resolved differential conductance also contains information about the topography of the sample. As explained in the revised manuscript in this representation the structural boundaries of the islands can be identified more easily, and thus also it becomes evident where the domain walls are located within the RhCo islands.

3) I guess the difference between upper panel and lower one is the spatial position of the dI/dV maps. Is this correct? This point is not clearly mentioned in either text or caption. Initially, I thought that these correspond to dI/dV maps taken with different condition for a single position.

We thank the reviewer for pointing this out. In the revised version the positions of the two different areas (now Fig. 2) are indicated in the newly added Fig. 1.

4) I guess the magnetization direction of Cr tip is perpendicular to the film plane. Is this correct? Then, why do the domain walls show brighter contrast than the domain itself?

The tip magnetization direction is canted, i.e. it is sensitive to both out-of-plane as well as in-plane sample magnetization components. To make this clear we have added a side view sketch of the tip magnetization direction in current Fig. 2. In addition we have added the separate contributions of out-of-plane TMR, in-plane TMR and NCMR for the differential conductance signal in Supplementary Fig. S6.

5) DFT calculation indicates that the existence of Rh layer changes the q -dependence of $E(q)$ qualitatively from q^2 to q^4 , but why does this occur? The result is fitted with atomistic spin model, but the original system is a metallic one. How is this understood?

The change of the exchange interactions within the Co film, which is reflected in the energy dispersion $E(q)$, is caused by the hybridization of $3d$ - and $5d$ -states at the Co/Ir and of $3d$ - and $4d$ -states Rh/Co interfaces. The impact of such hybridization has been studied before e.g. for Fe monolayers on different $4d$ - and $5d$ -transition-metal surfaces (B. Hardrat *et al.*, PRB **79**, 094411 (2009)) or for Pd or Rh overlayers on Fe/Ir(111) (B. Dupé *et al.*, Nature Comm. **5**, 4030 (2014) and N. Romming *et al.*, PRL **120**, 207201 (2018)). As a general trend, it was found that with reduced band filling of the adjacent $4d$ -/ $5d$ -layer the nearest-neighbour exchange interaction J_1 is reduced and can even change sign from ferro- to antiferromagnetic in Fe films (B. Hardrat *et al.*, PRB **79**, 094411 (2009)).

If J_1 is reduced long-range RKKY exchange interactions typical for itinerant magnets can play an important role for the magnetic ground state. The interplay of exchange beyond nearest neighbours and DMI can lead to complex noncollinear ground states such as spin spirals. E.g. Rh/Fe/Ir(111) exhibits a deep energy minimum for spin spirals of about 17 meV/Fe atom driven by exchange interactions while Pd/Fe/Ir(111) has a shallow spin spiral energy minimum due to DMI which allows for skyrmions in a magnetic field.

For Co films the interface hybridization has a similar effect on the exchange interactions, but not quite as large. Thus Rh/Co/Ir(111) possesses depending on the Rh stacking either a very small spin spiral minimum (0.7 meV/Co atom) or the ferromagnetic state is slightly more favourable. In Co/Ir(111) (Perini *et al.*, PRB **97**, 184425 (2018), Ref. [31]), for comparison, the Co/Ir hybridization reduces J_1 by about 30% with respect to Co/Pt(111) (Zimmermann *et al.*, arxiv:1904.06954, accepted for publication in PRB) but shows little exchange frustration and is a typical ferromagnetic film.

The second point of the referee concerns the applicability of an atomistic spin model for a metallic system such as Rh/Co/Ir(111). Due to strong localization of the $3d$ orbitals in transition-metals such as Co or Fe one can define a localized magnetic moment by integrating the spin density over an atomic volume, i.e. a sphere of radius of about 1 Å around every atomic site (see e.g. "Atomistic spin dynamics: foundations and applications" by O. Eriksson, A. Bergman, L. Bergqvist, J. Hellsvik, Oxford University Press, 2017). These magnetic moments are relatively unaffected in size by rotations of spin directions on different atomic sites. This allows to map the total energy of different noncollinear spin structures, e.g. spin spirals, to an effective Heisenberg-type spin model (see e.g. "Exchange interactions, spin waves, and transition temperatures in itinerant magnets", I. Turek, J. Kudrnovský, V. Drchal & P. Bruno, Philos. Mag. **86**, 1713 (2006)). However, due to the long-range RKKY-type exchange interactions in metals the description needs to go beyond nearest-neighbours, e.g. in our films up to 10 nearest-neighbour shells need to be taken into account to adequately model the energy dispersion.

Such an atomistic spin model with DFT parameters has been applied to calculate the Curie temperature of $3d$ transition metals (e.g. in M. Ležaić *et al.*, Appl. Phys. Lett. **90**, 082504 (2007) for Co and Fe with values of T_C which deviate from the experimental values by about 10%) or ultrathin films (e.g. the change of T_C of Co films on Pt(111) under applied electric fields: M. Oba *et al.*, Phys. Rev. Lett. **114**, 107202 (2015)). In the context of skyrmions for the ultrathin film system Pd/Fe/Ir(111) it has provided a good description of skyrmion properties such as their diameters, profiles or stability (S. von Malottki *et al.*, Sci. Rep. **7**, 12299 (2017),

P. F. Bessarab *et al.*, Sci. Rep. **8**, 3433 (2018)). Applications to other skyrmion film systems include Co/Ru(0001) (M. Hervé *et al.*, Nature Comm. **9**, 1015 (2018)).

In the revised version of the manuscript we have written an additional paragraph in the introduction which discusses the effect of exchange frustration in ultrathin films in order to include the information above also for the reader of the manuscript. We have also included a reference to the excellent introduction into basic concepts and applications of atomistic spin dynamics as given in the book: "Atomistic spin dynamics: foundations and applications" by O. Eriksson, A. Bergman, L. Bergqvist, J. Hellsvik, Oxford University Press, 2017.

Reply to Reviewer #3:

This study entitled "Isolated zero field sub-10 nm skyrmions in ultrathin Co films" by S. Meyer et al is part of the large research effort currently made in solid state physics on the investigation of magnetic skyrmions both for addressing fundamental questions but also for the potential use of these topologically protected magnetic textures in novel spintronic conceptual devices. These developments require to find solutions to stabilize extremely small isolated skyrmions. One of the important results of this study is precisely to have found such a case in the thin film system Co/Rh on Ir (111). Indeed, combining advanced magnetic imaging by STM and DFT calculations, the authors bring clear evidences of the impact of exchange frustration for the stabilization of 5-nm diameter isolated skyrmions. Moreover, the balance between all the magnetic interactions in this system makes that these ultra-small skyrmions can be stable even at zero applied field. Even if this observation has been done at low temperature, this result provides a clear strategy to achieve similar results at room temperature in multilayer systems. For this reason, I believe that this work is of interest for the growing community of researchers working on skyrmions and more generally nanomagnetism. In conclusion, I recommend this work for publication in Nature Communications after having clarified a few points that are listed hereafter.

We thank the reviewer for the positive judgement on the impact of our work for the community of researchers on nanomagnetism and for the issues which the referee raised. We believe that the manuscript in its revised form has very much benefitted from the changes triggered by the reviewer's comments and hope that the referee agrees. In the point-by-point reply below we address all the remaining issues raised in the referee report.

1) The most important point concerns the comparison between experiments and DFT calculations made in page 3. As far as I understand, it comes out from the calculations that for Rh hcp on Co, the ground state is not the one observed by STM. Then the authors comment this discrepancy only by saying that the DMI is typically overestimated using their approach! It would be important that they bring some further explanations. How much do they have to reduce DMI in order to "agree" with the experimental observation? In the present version, it is not clear whether the DMI overestimation is specific to the Rh/Co system or if it is more general result.

The referee is right in saying that the magnetic ground state predicted from our DFT calculations for the hcp stacking of the Rh overlayer on Co/Ir(111) is different, i.e. a spin spiral, from that observed in the experimental system, i.e. a ferromagnetic state. However, the spin spiral energy minimum is extremely small – only 0.7 meV/Co atom below the ferromagnetic state. For fcc-Rh stacking, we find the ferromagnetic state to be slightly more favourable than spin spirals. For comparison, the spin spiral minimum of fcc-Rh/Fe/Ir(111) obtained in DFT calculations is 17 meV/Fe atom below the ferromagnetic state and is driven by very strong frustration of exchange interactions. The spin spiral ground state was confirmed experimentally by SP-STM (Romming *et al.*, PRL **120**, 207201 (2018)).

We attribute the discrepancy between the DFT result and the experiment to three issues:

- The first issue is intermixing at the Rh/Co interface observed in the experimentally prepared films (see also point 2 below raised by this referee). Such intermixing has recently been studied in detail by a DFT approach (KKR method) which allows for an efficient treatment of such disorder. In particular it was found that at Co/Pt interfaces the DMI is reduced by 20% already upon a small amount of intermixing of 10-20% (Zimmermann *et al.*, APL **113**, 232403 (2019)). While the FLAPW and the KKR method give consistent results for the magnetic interactions (exchange, DMI) in Co/Pt(111) (as discussed in Zimmermann *et al.*, arxiv:1904.06954, accepted in PRB) our code does not allow to treat arbitrary amounts of intermixing. Therefore, we have tested the effect of intermixing by calculating spin spiral energy dispersions of Co/Rh/Ir(111), i.e. 100% intermixing (Fig. S8 of the revised supplementary information). We find that the DMI is not only reduced but that its sign is even reversed. This shows that intermixing can significantly reduce the DMI of these Co films. The exchange interaction will also become a little less frustrated due to intermixing (Fig. S8) which will also promote the ferromagnetic state with respect to the spin spiral energy minimum.
- As stated by the referee there can also be an overestimation of the DM interaction by using first-order perturbation theory with respect to spin spirals compared to a self-consistent calculation in a supercell. For an Fe/Ir bilayer we found a deviation of 10-20% (Meyer *et al.*, PRB **96**, 094408 (2017)) in the energy contribution due to DMI and for Co/Pt(111) a deviation of about 25% was found (Zimmermann *et al.*, arxiv:1904.06954, accepted for publication in PRB). Since the self-consistent method relies on supercell calculations it restricts the possible spin structures we can take into account to very short spin spirals. On the other hand, first-order perturbation theory can be applied to spin spirals of arbitrary spiral vector q , i.e. arbitrary spin spiral length.
- Finally, we would like to mention that the DFT and spin dynamics studies on Rh/Co/Ir(111) were performed *prior* to the experiments and initiated them. The starting point was Co/Ir(111) – a system with a ferromagnetic ground state and chiral domain walls – for which SP-STM and DFT calculations give a consistent picture (Perini, Meyer, *et al.*, PRB **97**, 184425 (2018), Ref. [31]). The frustration of exchange interactions was anticipated from the difference between Co/Ir(111) and Co/Pt(111), the later has a much larger nearest-neighbour exchange constant, and the system

Rh/Fe/Ir(111) (Romming *et al.*, PRL **120**, 207201 (2018)) which demonstrated how exchange frustration can occur due to hybridization in Rh/3d/Ir systems. The DFT calculations show that our educated guess was right and that Co/Ir(111) and Rh/Co/Ir(111) behave *qualitatively* very different (cf. Fig. 4 of our paper). Small deviations with respect to absolute numbers, i.e. the depth of the spin spiral energy minimum, are however within the accuracy that can be expected from the exchange-correlation functionals.

In conclusion, we have reduced the DMI in our atomistic spin dynamics simulations for hcp-Rh/Co/Ir(111) by 50% in order to model the experimental system as stated in the caption to Supplementary Table S11 of the revised Supplementary Information. We had briefly mentioned the issues (i) and (ii) in the previous version of the manuscript and in the supplementary information. However, to make this point clearer in the manuscript we have extended the discussion in the main text to a full paragraph of the revised manuscript and also cite the DFT work on Co/Pt(111) (Zimmermann *et al.*, arxiv:1904.06954, accepted in PRB) which has not been available on the submission of our manuscript.

2) In the same paragraph, the authors propose that additional reduction of the actual DMI value can come from intermixing. Did they try to include some intermixing in the DFT calculations and if yes, how large is the intermixing to be introduced in order to agree with experiments?

See our answer to the previous question.

3) As the possible intermixing between Rh and Co is also invoked to explain the shape of the DWs, I wonder if the STM experiments cannot be used to extract some more quantitative information on the actual intermixing?

Unfortunately the STM experiments do not reveal quantitative information on the degree of intermixing in the RhCo bilayer. Neither from the constant-current topography images nor from the maps of differential conductance (see new Suppl. Fig. S2 for respective data) we are able to identify the different atomic species, possibly due to their similar electronic properties as isoelectronic elements.

4) It is not clear to me what are the reasons to show two different regions in Fig1a and 1b and why it brings more information than showing only one?

The Reviewer is correct, that showing the two different regions does not bring additional information, in other words the two different RhCo island show similar behaviour. We are showing both of them to emphasize that zero-field magnetic skyrmions with opposite magnetizations coexist in the virgin state of this sample system. We also analyze both of these opposite skyrmions in panels (c) and (d) (now Fig. 2).

In Fig 1c, it is rather clear that the skyrmion profile is not symmetric. How such a non-symmetric skyrmion can be obtained in the STM simulations (green curve in Fig 1c)? Finally, in Fig. 1e, the statement made that the fitted profile shows the typical continuous rotation is not really justified, at least for the violet curve.

We have added Supplementary Fig. S6 with separate contributions of the out-of-plane TMR, the in-plane TMR, and the NCMR to explain the asymmetric shape observed with a canted tip magnetization. We have also included a side-view sketch of the tip magnetization direction in current Fig. 2.

5) For a non-specialist reader, it will be important to remind in the text the expression of the DW energy as it is discussed several times.

We have evaluated the DW energy given in the manuscript directly from our numerical atomistic spin dynamics simulations in order to include the effects from the exchange frustration explicitly. As discussed in the paper it is not possible to capture the energy dispersion in the vicinity of the ferromagnetic state within an effective nearest-neighbour exchange model. Therefore, we cannot apply the DW energy formula from the literature which is only known for the case in which the magnetic interactions can be adequately modelled by three parameters J , D , and K .

In the revised version of the manuscript we have modified the sentence on the DW energy in the following way to make this point clear:

"The DW energy obtained from our spin dynamics simulations with respect to the ferromagnetic state amounts to only 2.0 meV/nm, one order..."

6) Even if the authors mention that they focus on Rhhcp/Co on Ir(111), I would find interesting to discuss a bit more the other stacking Rhfcc/Co. For example, it is not clear if any STM experiments have been obtained on this stacking. If yes, it would be interesting to know if the predicted opposite rotation is observed?

We have now included the previous Suppl. Fig. S2 into the main text (current Fig. 1). It also shows several fcc-Rh/Co islands which are all single domain ferromagnetic. We are discussing this in the revised version.

7) Table S12: how do you understand the large frustration on DMI as D_3 is much larger than D_1 for example?

The DM interaction in transition-metal films is mediated by conduction electrons – as described e.g. in the Levy-Fert model (PRB **23**, 4667 (1981)) – which are also responsible for the long-range RKKY-type exchange interaction. Consequently, a term similar to that in the RKKY interaction, which oscillates as a function of separation between interacting spins, appears for the DMI. Therefore, it is not unexpected that the frustration occurs in both the

DMI and the exchange interaction. The relative strengths of the competing terms and the oscillation period depends on details of the electronic structure which is captured in DFT calculations.

For a Co monolayer on Pt(111), for example, the exchange interaction is dominated by the ferromagnetic nearest-neighbour term and the DMI can also be described well by a nearest-neighbour interaction (Zimmermann *et al.*, arxiv:1904.06954).

For a Co monolayer on Ir(111), on the other hand, we find a change of sign of the DMI contribution with the spin spiral vector q , i.e. as the spin spiral period decreases (Suppl. Fig. S7c). This leads to DMI constants with opposite signs.

For a sandwich system such as Rh/Co/Ir(111) there can also be a competition of contributions from the Rh overlayer and the underlying Ir substrate which can lead to DMI frustration. In Suppl. Fig. S7c the energy contribution due to spin-orbit coupling, i.e. the DMI, to the dispersion of spin spirals is shown. We see that in the vicinity of the $\bar{\Gamma}$ point the energy due to DMI depends linearly on the spin spiral vector q (up to $|q| \approx 0.1 \times 2\pi/a$, i.e. spin spiral periods of about 2.5 nm). In this regime the DMI can be captured by an effective nearest-neighbour DM constant (cf. Suppl. Table S11). However, at larger $|q|$, i.e. shorter spiral periods, we find a change of sign of the DMI contribution which is reflected in DMI constants with opposite sign.

REVIEWERS' COMMENTS:

Reviewer #1 (Remarks to the Author):

Reviewer 1

2nd Review of "Isolated zero field sub-10 nm skyrmions in ultrathin Co films" by Sebastian Meyer et al. submitted to Nature Communications.

Manuscript ID: NCOMMS-19-08562A

I have read the authors' response to my comments on their previous manuscript to find that they have solved my previous concerns thoroughly. Now that their revised manuscript is much improved, I feel rather positive on their manuscript for a publication in Nature Communications. So, I would like to make final suggestions to improve the clarity of their revised manuscript further just for readers' benefit.

Minor revisions (suggestions)

(1) I am afraid that readers get frustrated by the different orientations between Figure 1 and Figure 2. I suggest the authors to align Figure 1 with Figure 2.

(2) Judging from common concerns of the three reviewers ((3) of reviewer 2, (4) of reviewer 3, and reviewer 1 has the same opinion, too.), readers might feel uncomfortable by the implicit significance of presenting two different field-of-views in Figure 2a&2b. I think that the authors' intention to display the two field-of-views as the evidence of two opposite skyrmions should be described explicitly in the main text. Although it is described at the end of the first paragraph of page 4 as "It is quite remarkable, that these opposite magnetic skyrmions coexist in the virgin state of our Rh/Co film and that they do not collapse regardless of their small diameter.", it is not a bad idea to stress the significance by repeating several times in the main text.

(3) In addition, readers might feel uncomfortable because the two field-of-views are not labeled properly. I suggest using Greek numerals I and II to label the two field-of-views as indicated by the black boxes in Figure 1. Accordingly, use these labels at the top-left corners of the upper and lower panels of Figure 2a and those of Figure 2b. Revise the main text in accordance with these changes. For example, change "see Fig. 2a for a closer view of the sample areas indicated by the black boxes in Fig. 1." (line 14, page 3) as "see Fig. 2a for a closer view of the sample areas (I&II) indicated by the black boxes in Fig. 1."

End of Review

Reviewer #2 (Remarks to the Author):

I have read the authors' reply to my comments and revised manuscript to find that necessary revisions are appropriately made. I am totally satisfied with them and recommend publication of this article in Nature Communications.

Reviewer #3 (Remarks to the Author):

The authors have answered to all my comments as well as to all the other ones from the two other referees. Moreover, they have properly revised the manuscript. In consequence, I believe that the work deserves to be published in Nature Communications.

Reply to Reviewer #1:

I have read the authors' response to my comments on their previous manuscript to find that they have solved my previous concerns thoroughly. Now that their revised manuscript is much improved, I feel rather positive on their manuscript for a publication in Nature Communications. So, I would like to make final suggestions to improve the clarity of their revised manuscript further just for readers' benefit.

We thank the referee for reading our revised manuscript and our replies to the referee report, stating that our work is ready for a publication in Nature Communications after final minor suggestions. Below we address these suggestions by the referee point-by-point.

(1) I am afraid that readers get frustrated by the different orientations between Figure 1 and Figure 2. I suggest the authors to align Figure 1 with Figure 2.

Using the same perspective/rotation in both figures has other drawbacks, e.g. this would require more space in Fig. 2. With the new labelling suggested by the referee, the readers should have no problem identifying the two areas in Fig. 2, as they are currently presented.

(2) Judging from common concerns of the three reviewers ((3) of reviewer 2, (4) of reviewer 3, and reviewer 1 has the same opinion, too.), readers might feel uncomfortable by the implicit significance of presenting two different field-of-views in Figure 2a&2b. I think that the authors' intention to display the two field-of-views as the evidence of two opposite skyrmions should be described explicitly in the main text. Although it is described at the end of the first paragraph of page 4 as "It is quite remarkable, that these opposite magnetic skyrmions coexist in the virgin state of our Rh/Co film and that they do not collapse regardless of their small diameter.", it is not a bad idea to stress the significance by repeating several times in the main text.

With the two fields of view in Fig. 2, the data can be shown considerably enlarged compared to Fig. 1, while the relative position can be clearly seen in Fig. 1. We do not see a benefit of a single field of view in Fig. 1.

We thank the referee for his suggestion to stress the significance of observing zero-magnetic field skyrmions in the FM virgin state. We have restated this important result of our work in comparison to previous studies in the abstract:

"Here we report zero field isolated skyrmions **at $T = 4$ K with diameters below 5 nm emerging from the virgin ferromagnetic state** coexisting with 1 nm thin domain walls in Rh/Co atomic bilayers on Ir(111)."

and on page 5 of the revised manuscript at the end of the paragraph which starts with "In agreement with the experimental findings...". The last sentence now reads:

"Note that all previously found nanometre-sized isolated skyrmions in ultrathin films^{8,24,25,40} were induced by a magnetic field out of a spin spiral ground state **while in the Rh/Co films individual skyrmions exist in the virgin FM state at zero magnetic field.**"

The changes have been marked in bold face.

(3) In addition, readers might feel uncomfortable because the two field-of-views are not labelled properly. I suggest using Greek numerals I and II to label the two field-of-views as indicated by the black boxes in Figure 1. Accordingly, use these labels at the top-left corners of the upper and lower panels of Figure 2a and those of Figure 2b. Revise the main text in accordance with these changes. For example, change "see Fig. 2a for a closer view of the sample areas indicated by the black boxes in Fig. 1." (line 14, page 3) as "see Fig. 2a for a closer view of the sample areas (I&II) indicated by the black boxes in Fig. 1."

We agree with the referee and have used the labels I and II in Figs. 1 and 2 as suggested. In addition, we have made minor changes to the figures and their captions for clarity: Figs. 1 and 2 start now with "Perspective view of" and we added "colour gradient as in Fig.1" in both captions, following a request from the editor. The measurement values themselves are well-defined from the shown line sections. We also added "(arb. u.)" where it was missing in Fig. 2, and changed "x (nm)" to "distance (nm)" in Fig. 2 and 3.

Reply to Reviewer #2:

I have read the authors' reply to my comments and revised manuscript to find that necessary revisions are appropriately made. I am totally satisfied with them and recommend publication of this article in Nature Communications.

We thank the referee for reading our revised paper and answers to his/her questions and for recommending publication in Nature Communications.

Reply to Reviewer #3:

The authors have answered to all my comments as well as to all the other ones from the two other referees. Moreover, they have properly revised the manuscript. In consequence, I believe that the work deserves to be published in Nature Communications.

We thank the reviewer for reading our revised manuscript as well as our answers to the questions and comments of the referees and for the recommendation to publish it in Nature Communications.

Changes made in figures:

Fig.1: marked areas are labeled by "I" and "II"

Fig. 2: Labels "I" and "II" as in Fig. 1, "x(nm)" replaced by „distance (nm)“ and "(arb. u.)" added

Fig3: "x(nm)" replaced by „distance (nm)“

All changes in the manuscript text and captions are marked by the track changes feature.